# The role of survivin in the progression of pancreatic ductal adenocarcinoma (PDAC) and a novel survivin-targeted therapeutic for PDAC

**Matthew Brown[1], Wanbin Zhang[2], Deyue Yan[2], Rajath Kenath[3], Le Le[3], He Wang[3,4], Daniel Delitto[5], David Ostrov[6], Keith Robertson[7,8], Chen Liu[3,4]\*, Kien Pham[3,4]\***

**1** Department of Cell Biology and Neuroscience, School of Arts and Sciences, Rutgers University, Piscataway, New Jersey, United States of America, **2** School of Chemistry and Chemical Engineering, Shanghai Jiao Tong University, Shanghai, P.R. China, **4** Department of Pathology and Laboratory Medicine, Robert Wood Johnson Medical School, Rutgers University, New Brunswick, New Jersey, United States of America, **3** Department of Pathology and Laboratory Medicine, New Jersey Medical School, Rutgers University, Newark, New Jersey, United States of America, **5** Department of Surgery, Johns Hopkins University School of Medicine, Baltimore, Maryland, United States of America, **6** Department of Pathology, Immunology, and Laboratory Medicine, University of Florida, Gainesville, Florida, United States of America, **7** Department of Molecular Pharmacology and Experimental Therapeutics, Mayo Clinic, Rochester, Minnesota, United States of America, **8** Center for Individualized Medicine, Mayo Clinic, Rochester, Minnesota, United States of America

\* kien.pham@rutgers.edu (KP); chen.liu@rutgers.edu (CL)

**Data Availability Statement:** All relevant data are within the manuscript.

**Funding:** The authors received no specific funding for this work.

## Abstract

Treating pancreatic ductal adenocarcinoma (PDAC) remains a major hurdle in the field of oncology. Less than half of patients respond to frontline chemotherapy and the pancreatic tumor microenvironment limits the efficacy of immunotherapeutic approaches. Targeted therapies could serve as effective treatments to enhance the clinical response rate. One potential therapeutic target is survivin, a protein that is normally expressed during embryonic and fetal development and has a critical impact on cell cycle control and apoptosis. In adulthood, survivin is not present in most normal adult cells, but is significantly re-expressed in tumor tissues. In PDAC, elevated survivin expression is correlated with treatment resistance and lower patient survival, although the underlying mechanisms of survivin's action in this type of cancer is poorly understood. Using patient derived xenografts of PDAC and their corresponding primary pancreatic cancer lines (PPCL-46 and PPCL-LM1) possessing increased expression of survivin, we aimed to evaluate the therapeutic response of a novel survivin inhibitor, UFSHR, with respect to survivin expression and the tumorigenic characteristics of PDAC. Cell viability and apoptosis analyses revealed that repressing survivin expression by UFSHR or YM155, a well-known inhibitor of survivin, in PPCLs effectively reduces cell proliferation by inducing apoptosis. Tumor cell migration was also hindered following treatment with YM155 and UFSHR. In addition, both survivin inhibitors, particularly UFSHR, effectively reduced progression of PPCL-46 and PPCL-LM1 tumors, when compared to the untreated cohort. Overall, this study provides solid evidence to support the critical role of survivin in PDAC progression and proposes a novel survivin inhibitor UFSHR that can become an alternative strategy for this type of cancer.

**Competing interests:** The authors have declared that no competing interests exist.

## Introduction

Selecting effective therapeutic strategies for pancreatic ductal adenocarcinoma (PDAC) continue to be a major challenge in modern oncology. Of the over 50,000 new cases of PDAC expected to manifest annually, it is estimated that less than 10% of patients will survive more than 5 years [1]. Gemcitabine, a chemotherapeutic agent that analogs a pyrimidine nucleoside, has received FDA-approval as a first-line treatment for advanced pancreatic adenocarcinoma since 1996, yet its effectiveness, particularly in advanced stages, remains controversial. In an attempt to improve the efficacy of chemotherapy for advanced PDAC, many phase III clinical trials of combination strategies have emerged [2]. Two such combination regimens are FOL-FIRINOX (a cocktail of folinic acid (calcium folinate), fluorouracil, irinotecan, and oxaliplatin) and gemcitabine in combination with albumin-bound paclitaxel [3]. However, less than 30% of patients see a response following treatment with either therapy [3]. Likewise, therapies targeting immune checkpoints such as PD-1 and CTLA-4 have also seen minimal success in PDAC. This is primarily due to a reduced antigenicity of the tumor itself and an immunosuppressive microenvironment rich with tumor-infiltrated desmoplastic stromal elements [3]. Therefore, targeted therapy could present a major opportunity to treat highly heterogenous pancreatic tumors that do not respond to existing chemotherapeutic agents or immunotherapies. One potential and attractive molecular target is survivin, a member of the Inhibitor of Apoptosis Protein (IAP) family.

Survivin is a 142-animo acid peptide that is encoded by Baculoviral IAP Repeat Containing 5 (BIRC5) gene located on chromosome 17 and usually exists in the form of a homodimer [4]. In addition to the full-length wild type (WT) transcript, there are six splice variants, a consequence of alternative splicing, that have been reported [5]. Physiologically, survivin has been shown to play a critical role in apoptosis; however, only the final protein products of the WT survivin, survivin ΔEx3, and survivin 3B transcripts have been confirmed to retain antiapoptotic function [6].

The survivin protein is actively expressed during embryonic stages of development but not in normal cell populations of adults, except for hematopoietic stem cells and vascular endothelial cells [7]. However, in many types of cancer including PDAC [8], this protein is significantly overexpressed when compared to that of adjacent normal tissue [4]. Since its expression is largely limited to neoplastic tissues, targeting survivin would therefore seek to minimize toxicity to noncancerous cells. In PDAC and other malignancies, survivin expression is inversely correlated with overall survival [9]. Clinical studies have shown that elevated survivin expression is also correlated with higher rates of recurrence [10] and treatment resistance [11].

Survivin acts on both extrinsic and intrinsic apoptotic pathways. Survivin regulation through the intrinsic apoptotic pathway is accomplished by stabilizing the activated X-linked inhibitor of apoptosis protein (XIAP) and thereby inhibiting caspase-3 and caspase-9 [11]. In the intrinsic pathway, Smac (second mitochondria-derived activator of caspases) negatively regulates survivin by preventing its interaction with downstream caspases [12]. In addition, survivin inhibits caspase-8 activity under the death ligand-dependent extrinsic pathway [11]. Survivin also plays a critical role in cell cycle progression. During mitosis, it promotes mitotic activity by aiding in the attachment of the microtubules to the kinetochore and by forming a chromosome passenger complex so that the chromosomes can properly align during mitosis [13]. Several studies have shown an alteration of survivin expression in different stages of the cell cycle, in which its level peaks during M-phase and significantly decrease in the G1 phase [14], [15]. The reduction of survivin in G1 phase may be due to the inhibition of cyclin-dependent kinase 4 (CDK4) [16] or to the binding of retinoblastoma protein (RB) at the survivin promoter [17], [18].

Survivin function is regulated at three known phosphorylation sites. At the threonine 34 residue, cyclin-dependent kinase 1 phosphorylates the survivin protein in order to promote interaction with the mitotic spindle and also to inhibit the activity of caspase-8 [19]. Aurora kinase B phosphorylates the threonine 117 residue to promote the attachment of the kinetochore to the microtubule [20]. Finally, protein kinase A phosphorylates the serine 20 residue to destabilize the survivin-XIAP complex, effectively inhibiting the anti-apoptotic function of survivin [21].

In oncogenesis, the aberrant overexpression of survivin has been linked to several causes. Neuroblastoma models have shown that one cause of survivin overexpression is BIRC5 gene amplification at 17q25 [22]. In ovarian cancer, heightened survivin expression was linked to demethylation of BIRC5's exon 1 [23] and dysregulation of the p53 tumor suppressor [24]. Finally, studies focused in colon cancer have shown that survivin becomes overexpressed following activation of the WNT-ß-catenin signaling cascade [25]. At the phenotypic level, the overexpression of survivin has been linked to tumor progression in prostate cancer [26], enhanced migratory potential of gastric cancer cells [27], and increased survival and chemoresistance in breast cancer models [28].

A survivin inhibitor, YM155 (sepantronium bromide), has been developed to target solid tumors and blood cancers that overexpress survivin. The mechanism of YM155-induced survivin repression has been cited in the past as principally acting through transcriptional inhibition. It has been hypothesized that YM155 destabilizes the transcriptional complex at the BIRC5 promoter by binding the transcription factors Interleukin Enhancer-binding Factor 3/NF110 [29] and SP1 [30]. However, while YM155 does in fact reduce survivin mRNA levels and this likely contributes to the reduction seen in survivin protein concentration, more recent studies have shown that the mechanism whereby YM155 downregulates survivin protein expression seems to be primarily due to changes in complex translational control cascades.

Polysome profiles were generated in renal and prostate cancer cell lines following YM155 treatment which showed that YM155 substantially suppresses cap-dependent translation of mRNAs that include survivin [31]. Before any noticeable changes in mRNA transcripts, this experiment showed a significant suppression in translation as early as 4 hours after the start of YM155 treatment. This study also showed that YM155 can change the phosphorylation status of known mTOR-target proteins that regulate translation (i.e. ribosomal protein S6 (rS6) and 4E-BP1). The last critical evidence from this study revealed that when cells were treated with MG132 (a proteasome inhibitor), there was no significant reversal of survivin loss by YM155, indicating that YM155 likely does not act through a proteasomal mechanism. YM155 has been evaluated clinically and has reached phase II trials for the treatment of melanoma and non-small cell lung cancer [32]. However, in addition, to a variable treatment response clinically, there is concern that the lack of specificity of YM155 gives rise to off-target effects such as increasing the expression of BIRC8, a pro-survival gene [33].

The goal of this study was to evaluate the therapeutic value of a novel survivin inhibitor that can be used as an alternative option to target pancreatic tumors that overexpress survivin. We examined the efficacy and efficiency of UFSHR, a novel synthetic derivative of YM155, in inhibiting survivin expression and in regulating both *in vitro* tumorigenic activities (including cell proliferation, apoptosis, migration) and *in vivo* tumor progression, using our primary pancreatic cancer lines (PPCLs) established from patient-derived tumor xenografts from primary PDAC tumors (PPCL-46) and hepatic metastases (PPCL-LM1) [34].

Unlike commercially available, highly passaged pancreatic cancer cell lines that have limited translational value, our well-characterized PDX-derived cancer lines serve as a strong preclinical model for their ability to efficiently preserve the tumor heterogeneity that is unique to each individual [34]. In this context, our PDXs appear as a more reliable model to characterize the

biology of PDAC and disease responsiveness to novel experimental treatments in precision medicine.

The data from this investigation showed that inhibiting survivin expression with either UFSHR or YM155 decreased *in vitro* cell proliferation by inducing apoptosis in PPCLs in a dose-dependent manner. *In vitro* wound healing analysis also showed that these survivin inhibitors effectively impaired the ability for PDAC cells to migrate. While the effectiveness of UFSHR on tumor functions is slightly less potent than YM155 in *in vitro* analysis, this novel survivin inhibitor significantly reduced survivin levels and halted the tumor progression on tumor-bearing mice, when compared to that of the animals treated with vehicle or YM155. Overall, this study re-confirms that survivin plays a critical role in the development of PDAC and that the newly developed survivin inhibitor UFSHR can become a robust therapeutic strategy for PDAC treatment.

## Materials & methods

### Animals

NOD-scid IL2Rγnull (NSG) mice were purchased from Jackson Laboratories. All procedures were carried out in accordance with the guidelines of the Institutional Animal Care and Use Committee (IACUC). The animal study performed in this project was approved by Rutgers IACUC (protocol number PROTO999900191).

### Cell lines and culture conditions

Primary pancreatic cancer lines (PPCLs) were established from patient-derived xenograft (PDX) tumors in our laboratory at the University of Florida, using the protocol previously described [34]. Informed written consent was obtained from all patients, and the collection of all patient material was approved by the University of Florida Institutional Review Board. Cell lines used in this study were PPCL-46 and PPCL-LM1. PPCL-46 was generated from a PDX tumor specimen derived from the primary lesion of a 75-year-old female patient with stage III PDAC (T3N2M0, 8th AJCC edition) collected on 05/2014. PPCL-LM1 was generated from a PDX tumor specimen derived from a hepatic metastatic lesion of a 65-year-old male patient with stage IV PDAC collected on 06/2013. Cell lines were maintained in advanced Dulbecco's Modified Eagle Medium with nutrient mixture F12 (Gibco, Gaithersburg, MD), 10% fetal bovine serum (Life Tech Cat No. 10082147), 6 mM GlutaMax (Life Tech Cat No. 35050061), 1% Pen/Strep (Life Tech Cat No. 15140122), 1 μM Hydrocortisone (Sigma Cat No. H6909), 200 nM Dexamethasone (Sigma Cat No. D2915), and 10/0.25 μg/mL Gentamicin/Amphotericin (Fisher Cat No. R01510). Cells were grown in a humidified incubator at 37°C with 5% $CO_2$. All cultured cells used in these experiments were kept at low passage number (less than 20).

### Immunoblotting analysis

Cells were seeded in 6-well plates at a concentration of 230,000 cells/mL in 3 mL of media. Cells were treated with 10 nM YM155 and 100 nM UFSHR for 24 and 48 hours. Protein was harvested using RIPA Lysis and Extraction Buffer (ThermoFisher Scientific Cat No. 89900) and 1X protease (Sigma-Aldrich #P8340) and phosphatase inhibitor (Sigma-Aldrich #P5726 and #P0044) cocktails. Protein concentration was calculated by measuring the absorbance of the sample in the presence of Bradford reagent. Lysates were then treated with 4X Laemmli sample buffer and 2-mercaptoethanol and heated at 90°C for 10 minutes before being stored at -20°C. 30 μg of samples were separated on 4–20% SDS-PAGE gels and then transferred

onto PVDF membranes. The membranes were then incubated with blocking solution (5% bovine serum albumin, 1X TBS, and 0.1% Tween-20) for 60 minutes. After blocking, primary monoclonal antibodies against survivin (Rabbit monoclonal antibody, Cat No. NB500-201, Novus Biologicals, Centennial CO. Antibody Registry ID: AB_10001517) or alpha-actin (Mouse monoclonal antibody, Cat No. sc-130617, Santa Cruz Biotechnology, Dallas TX. Antibody Registry ID: AB_1563153) were applied and left to incubate overnight at 4˚C. Membranes were then incubated with HRP-conjugated anti-mouse (Cat No. 7076S. Cell Signaling, Centennial CO. Antibody Registry ID: AB_330924) or anti-rabbit secondary antibodies (Cat No. 7074P2, Cell Signaling, Centennial CO. Antibody Registry ID: AB_2099233) for 1 hour. A dilution factor of 1:1000 was used for all antibodies with the exception of anti- alpha actin where a 1:20000 dilution factor was used. SuperSignal West Femto and SuperSignal West Pico substrates were used for detection.

### Reverse transcriptase polymerase chain reaction (RT-PCR)

Cells were treated with 10 nM YM155 or 100 nM UFSHR in a 6-well plate for 24 or 48 hours. TRIzol™ reagent (ThermoFisher Scientific, Waltham MA) was used to extract total RNA from the cells. 1 μg of RNA was reverse-transcribed using Maxima™ First Strand cDNA Synthesis Kit (ThermoFisher Scientific, Waltham MA). PCR amplification was performed using Applied Biosystems PowerUp™ SYRB™ Green Master Mix (ThermoFisher Scientific, Waltham MA) for quantitative real-time PCR. The PCR primers were: BIRC5 forward: 5'–CTGCTGTGGACCC TACTG–3'; reverse: 5'– AACTGCGTCTCTGCCAGGAC–3'; GAPDH forward: 5'– ACAA CTTTGGTATCGTGGAAGG–3', reverse: 5'– GCCATCACGCCACAGTTTC–3'. The relative expression of survivin mRNAs were quantitated using the relative quantitation method. Each experiment was independently repeated at least two times.

### Flow cytometry

Cells were trypsinized with Accutase solution, washed with ice-cold PBS, and then incubated with FITC-conjugate primary antibody anti-survivin (Dilution factor 1:10, Cat No. 130-106-740, Miltenyi Biotec, Somerville MA. Antibody Registry ID: AB_2653604). The stained cells were analyzed by flow cytometry, using the BD LSR II system (BD Biosciences, San Diego CA). The detection of apoptosis was performed with FITC Annexin V Apoptosis Detection Kit I (BD Biosciences, San Diego CA), according to manufacturer's protocol. Briefly, cells were dissociated with Accutase solution, washed with ice-cold PBS, re-suspended at the concentration of $1 \times 10^5$ cells in 100 μL of binding buffer, and stained with FITC Annexin V and Proprium Iodide at room temperature with light protection. After 15 minutes, cells were directly analyzed with a FC 500 flow cytometer system (BD Biosciences, San Diego CA). All data were further analyzed with FlowJo software version 8 (Tree Star). Each experiment was repeated at least two times.

### Cell proliferation assay

Cells were seeded in 96-well plates at a density of 250,000 cells/mL in 100 uL of the medium described in the Cell Culture section. Each line was treated with the survivin inhibitor YM155 or UFSHR (Selleck Chemicals, Houston TX) at concentrations of 1000 nM, 100 nM, 10 nM, 1nM, and 0.1 nM. 24 and 48 hours after treatment, cell viability was evaluated with Cell Counting Kit-8 assay (Dojindo Molecular Technologies Inc., Rockville MD), according to the manufacturer's protocol. The absorbance of yellow formazan dye, a derivative of water-soluble [2-(2-methox y-4-nitrophenyl)-3-(4-nitrophenyl)-5-(2,4-disulfophenyl)-2H-tetrazolium] monosodium salt from dehydrogenase activity in cells was detected with colorimetric microplate reader

at 450 nm. The $IC_{50}$ for each experimental group was calculated with GraphPad Prism software version 7 (P<0.0001). This assay was performed in triplicate and was repeated three times.

### Wound healing assay

In vitro cell migration was evaluated with a CytoSelect 24-well would healing assay (Cell Biolabs Inc., San Diego CA). Briefly, cells were seeded at a density of 350,000 cells/mL in 2mL/well of a 24-well plate containing a wound healing insert placed in the center of each well. After 24 hours, the inserts were removed and cells were washed with PBS. Then each cell line was treated with YM155 or UFSHR at concentrations of 5 nM and 50 nM respectively. After 24-hour or 48-hour time points, images were taken with an inverted microscope. The effect on migration was quantified by the unoccupied area, measured in square pixels using ImageJ software. Each experiment was performed in triplicate and repeated at least two times using different cell preparations.

### *In vivo* tumorigenicity and drug treatment

To establish tumorigenicity of cultured primary PPCLs, $3 \times 10^6$ cells from each patient's cell population was suspended in 200 μL of a 1:1 mixture of Dulbecco's Modified Eagle Medium with nutrient mixture F12 and Matrigel Matrix (Corning, Corning, NY) and inoculated subcutaneously into the right flank of 8-week-old nonobese diabetic NSG mice (The Jackson Laboratory, Bar Harbor, ME). Mice were inspected twice a week, and tumor size was measured with a digital caliper. After a palpable growth of the tumor was confirmed, the animals were given either YM155 or UFSHR at a dosage of 0.01 mg/kg and 0.1 mg/kg respectively diluted in sterile PBS, once a day for five days per week for four weeks via intraperitoneal injection. A control group of mice was treated with vehicle only. There were 6 animals per treatment group. Body weight and tumor size were measured once a week. All animals were euthanized one week after the last treatment. At this endpoint, tumors were collected, fixed in 10% formalin and subjected to histologic analysis with hematoxylin and eosin staining and immunohistochemical analysis. The percentage of survivin expression in the tumor was examined and quantitated by an experienced pathologist. The liver, kidneys, and spleens were also collected for toxicity assessment of the treatment. The tumor size was used to evaluate the growth response to the treatment. The response curve was statistically analyzed with GraphPad Prism software version 7. All procedures were carried out in accordance with the guidelines of the Rutgers University Institutional Animal Care and Use Committee.

### Statistical analysis

Statistical analyses were performed with GraphPad Prism software version 6. Data are presented as means and standard error of mean (SEM). Differences between groups in *in vitro* analyses were calculated with the Student's t-test or two-way ANOVA followed by the Student's t-test. The animal number used in the *in vivo* study was justified with Power and Sample Size Calculation software version 3.1.6 (Vanderbilt University) with a power of 90% at a significant level of 5%. For all experiments, significance is denoted by * (P<0.05), ** (P<0.01), *** (P<0.001), or **** (P<0.0001) as indicated in the figures and legends.

## Results

### Survivin is overexpressed in pancreatic adenocarcinoma and patient-derived xenograft models of PDAC

The expression of survivin is well documented in various types of cancer, including tumors of the colon, prostate, lung, breast, and liver. In pancreatic cancer, 68% of tumor tissues are

positive for survivin expression, when compared to normal tissues [35] (Fig 1A). In addition, an analysis to study the correlation between survivin expression and clinicopathologic data from 170 patients from the TCGA database revealed that the overexpression of survivin is inversely correlated with the patient survival rate (Fig 1B). A significantly shorter mean survival time was evident in pancreatic tumors having high expression of survivin, as compared to those with low levels. This analysis also shows that survivin is an independent variable that correlated with overall survival (P = 0.0004648). In our patient-derived xenograft models of pancreatic cancers, the overexpression of survivin is detected not only in PDX tumor tissues, using immunohistochemical analysis (Fig 1C), but also in corresponding pancreatic cancer cell lines (PPCLs), using flow cytometric analysis. (Fig 1D). The human pancreatic ductal epithelial (HPDE) cell line is negative for this protein. These observations suggest that survivin may be a potential therapeutic target for treating pancreatic cancer, and that our PPCLs developed from PDX tumors provide a suitable model to evaluate this hypothesis.

## UFSHR effectively reduces survivin expression in PPCLs

YM155 has been evaluated clinically to target solid tumors that overexpress survivin and this compound has reached Phase II clinical trials for the treatment of various types of blood and solid cancers, but these clinical investigations have not included advanced pancreatic adenocarcinoma [32]. In an attempt to discover new therapeutic options targeting survivin, we aimed to evaluate the efficacy and efficiency of a newly developed small molecule inhibitor on survivin suppression, 1H-Naphth[2,3-d]imidazolium,1,2-dimethyl-4,9-dioxo-3-[(4-phenylmethyl)methyl]-,iodide (1:1), namely UFSHR. This newly synthesized compound shares the same core 2-Methyl-1H-naphth[2,3-d]imidazole-4,9-dione with YM155's structure. However, UFSHR tethers to 1,4-dimethyl benzene while YM155 tethers to two functional groups including 2-methylpyrazine and methoxyethane (Fig 2A). The impact of UFSHR on survivin expression was evaluated in PDX-derived PDAC cell lines with western blot analysis at different dosages and time points. Our data showed that 24 hours after the treatment commenced, UFSHR effectively reduced the expression of survivin in a dose-dependent manner in both PPCL-46 and PPCL-LM1 cells. In terms of potency, the equivalent sensitivity of UFSHR and YM155 on survivin inhibition was achieved at 10 nM and 100 nM, respectively, suggesting that UFSHR is slightly less potent than YM155 at this time point (Fig 2B, top panel). At the 48-hour time point, UFSHR demonstrated a significant improvement on its potency toward survivin expression, especially in PPCL-LM1 (Fig 2B, middle panel). Analysis for mRNA expression showed that UFSHR contributed to the reduction of BIRC5 mRNA on PPCL-LM1, but not PPCL-46 cell line (Fig 2B, bottom panel). This finding suggests that the mechanism whereby UFSHR downregulates survivin protein expression may be different from that of YM155 and its effect on survivin mRNA expression may vary slightly depending on the unique molecular actors involved in each individual's tumor. Altogether, these observations strongly support the potential of this novel compound as another effective repressor of survivin protein expression.

## UFSHR reduces cell viability while inducing apoptosis in PPCLs

Survivin's ability to regulate cell survival makes it an attractive oncogene for targeted cancer treatment. We next examined this cancer-related phenotype of PPCLs with a serial dosage of UFSHR at different time points, using cell viability analysis (Fig 3A). After a 24-hour treatment, UFSHR showed an ability to inhibit cell viability in both the PPCL-46 and PPCL-LM1 models at an $IC_{50}$ of 52.7 nM and 50.67 nM, respectively while YM155 presented a better sensitivity in both lines (10.03 nM and 3.03 nM, respectively). While the sensitivity of both UFSHR

Figure 1

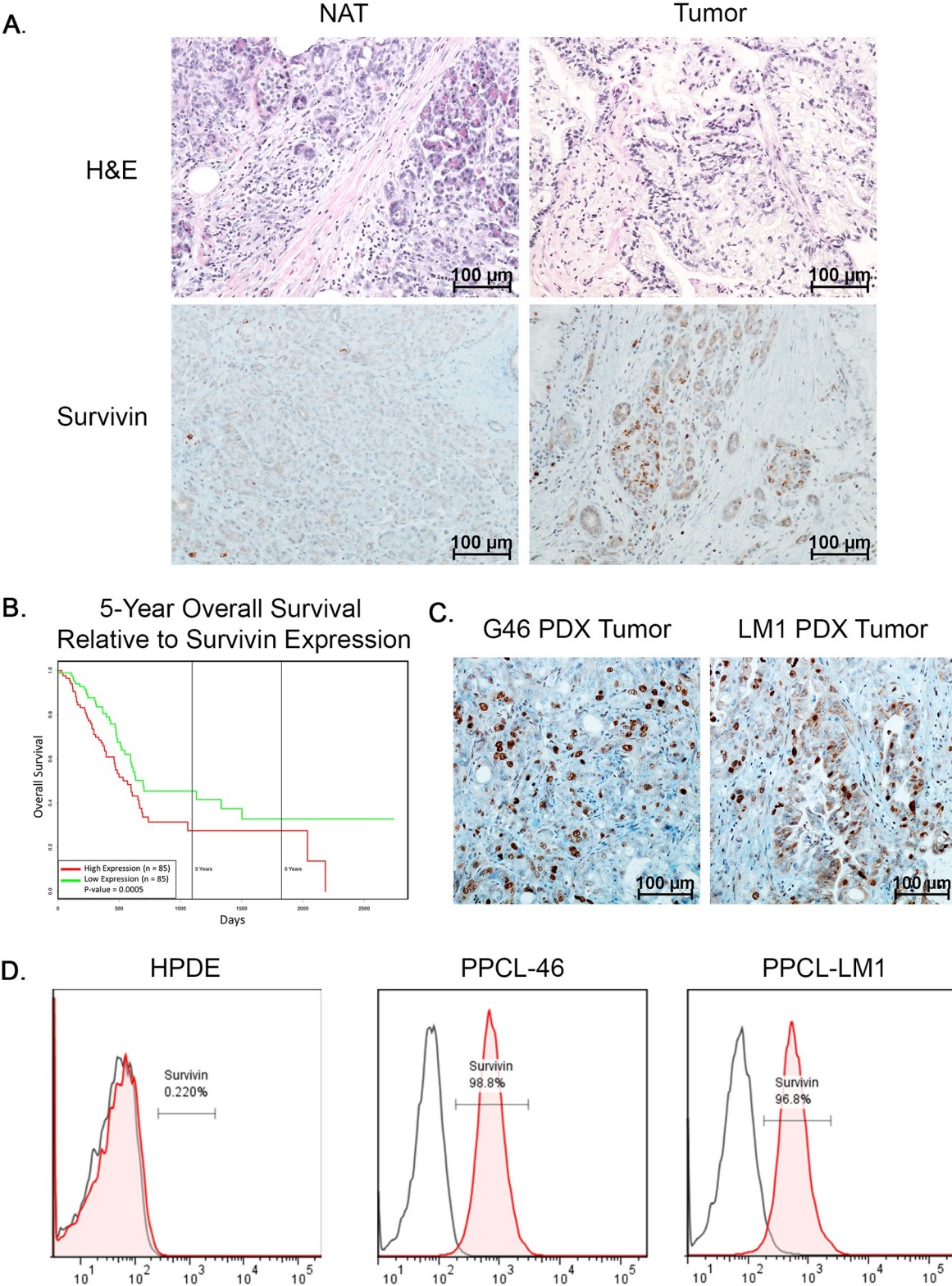

**Fig 1. Survivin is overexpressed in pancreatic adenocarcinoma and its expression is inversely correlated with patient survival.** (A) Survivin expression in a PC patient tumor and normal adjacent tissue. (B) Kaplan-meier survival curve comparing high and low survivin

expression from 185 PC patients from the TCGA database. High and low expression is relative to mean of gene expression. (C) Immunohistochemical staining for survivin in PDX tissue. (D) Flow cytometric analysis of survivin expression in HPDE cells and PPCLs.

and YM155 to cell viability remained unchanged after 48 hours in PPCL-LM1, there was a significant increase in the sensitivity to UFSHR in PPCL-46 with a nearly 17-fold change in potency. YM155 also increased its potency by a nearly 7-fold change with the longer treatment.

Consistent with the impact on cell viability, UFSHR, as well as YM155, also induced apoptosis in a similar manner, suggesting that the survivin inhibitors mediated reduction in cell viability as a consequence of the induction of apoptosis. Annexin V/propidium iodide assays confirmed that there was a significant reduction in the number of viable cells when PPCL-LM1 and PPCL-46 were treated with YM155 and UFSHR at both 24 and 48-hour time points (P<0.001) (Fig 3B). This coincided with a significant increase in the number of cells undergoing apoptosis in the cohort of cells treated with YM155 and UFSHR (P<0.001). Further analysis on subpopulations of apoptotic cells indicated a cell line-dependent difference in the

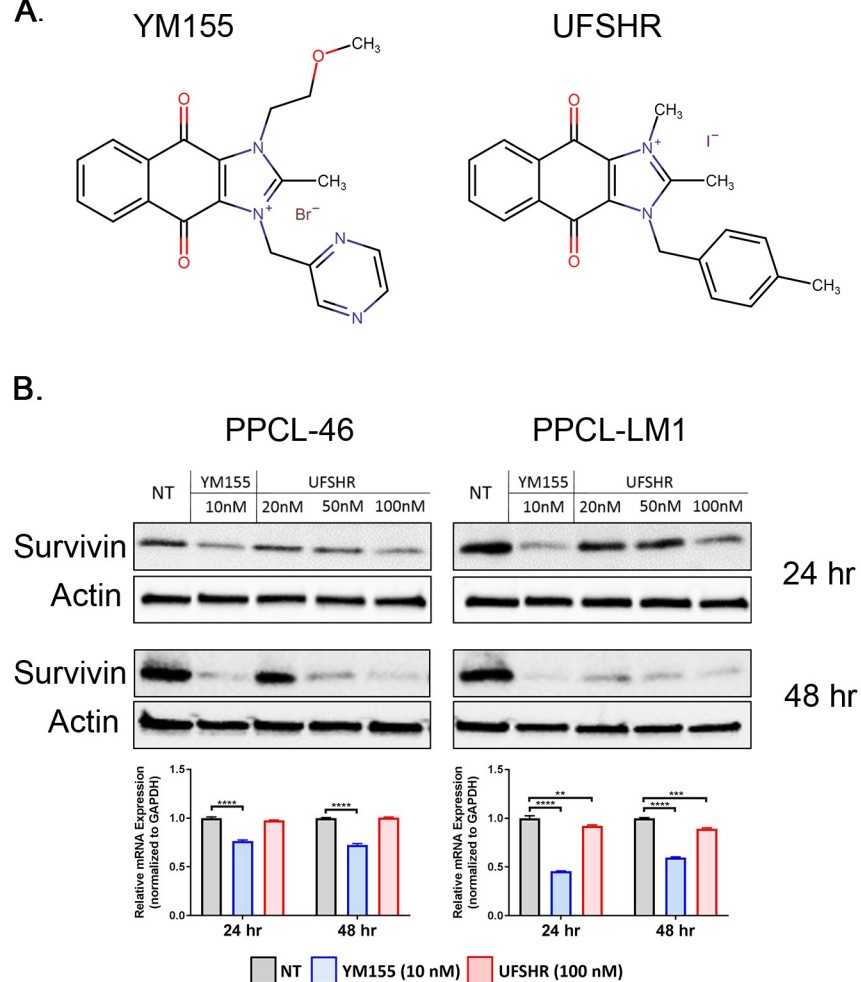

**Fig 2. Candidate compounds repress survivin expression in PPCLs.** (A) Structural formulas of YM155 and UFSHR. (B) Immunoblots of PPCL lysates collected 24 (top panel) and 48 (middle panel) hours after treatment commenced probing for survivin and alpha actin (loading control). Bottom panel: BIRC5 gene expression analyzed by RT-PCR.

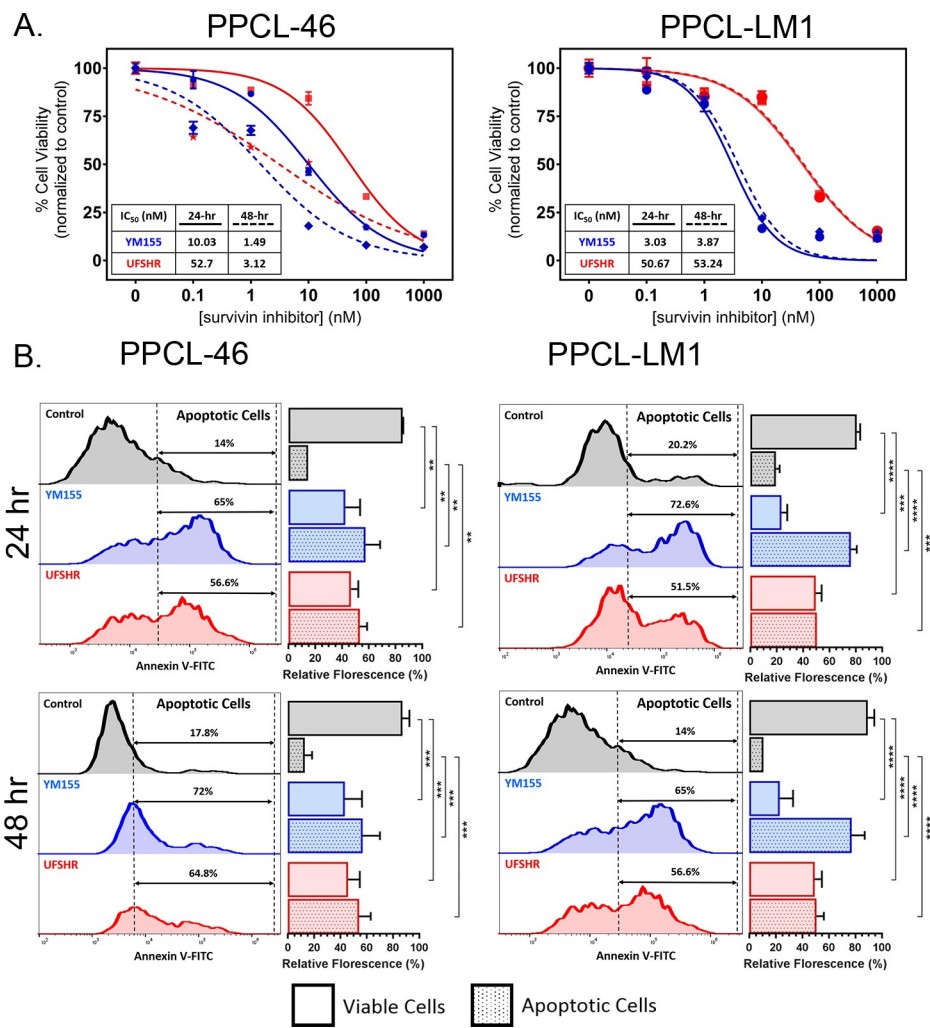

**Fig 3. Repressing survivin expression with YM155 and UFSHR reduces cell proliferation in PPCLs by inducing apoptosis.** (A) Cell viability assays following treatment with YM155 and UFSHR at 24 and 48-hour time points. (B) Annexin V/propidium iodide assays to identify apoptotic cell populations (V+) following treatment with YM155 and UFSHR at 24 and 48-hour time points. Open bar graphs indicate populations of viable cells. Filled bar graphs indicate populations of apoptotic cells. **P<0.01, ***P<0.001, ****P<0.0001.

distribution between early (Annexin V+/PI-) and late apoptotic cells (Annexin V+/PI+). This observation implies a possibility that survivin inhibitors may mediate cell apoptosis through a different mechanism between individuals (data not shown). Together these assays provide substantial evidence for the ability of UFSHR to control the proliferative phenotype of PPCLs. In addition, our data indicates that treatment duration also represents one of the key criteria in the determination of drug sensitivity for UFSHR.

## UFSHR reduces the ability for pancreatic tumor cells to migrate

The impact of UFSHR on the ability for tumor cells to migrate was evaluated with an *in vitro* would healing assay. Both UFSHR and YM155 significantly impaired the migration of both PPCL cell lines at a low concentration (Fig 4). At 48-hour time point, the migration of both PPCL-46 and PPCL-LM1 were significantly inhibited by either YM155 or UFSHR (P<0.001 and P<0.0001, respectively). Interestingly, the inhibitory effect of both survivin inhibitors was

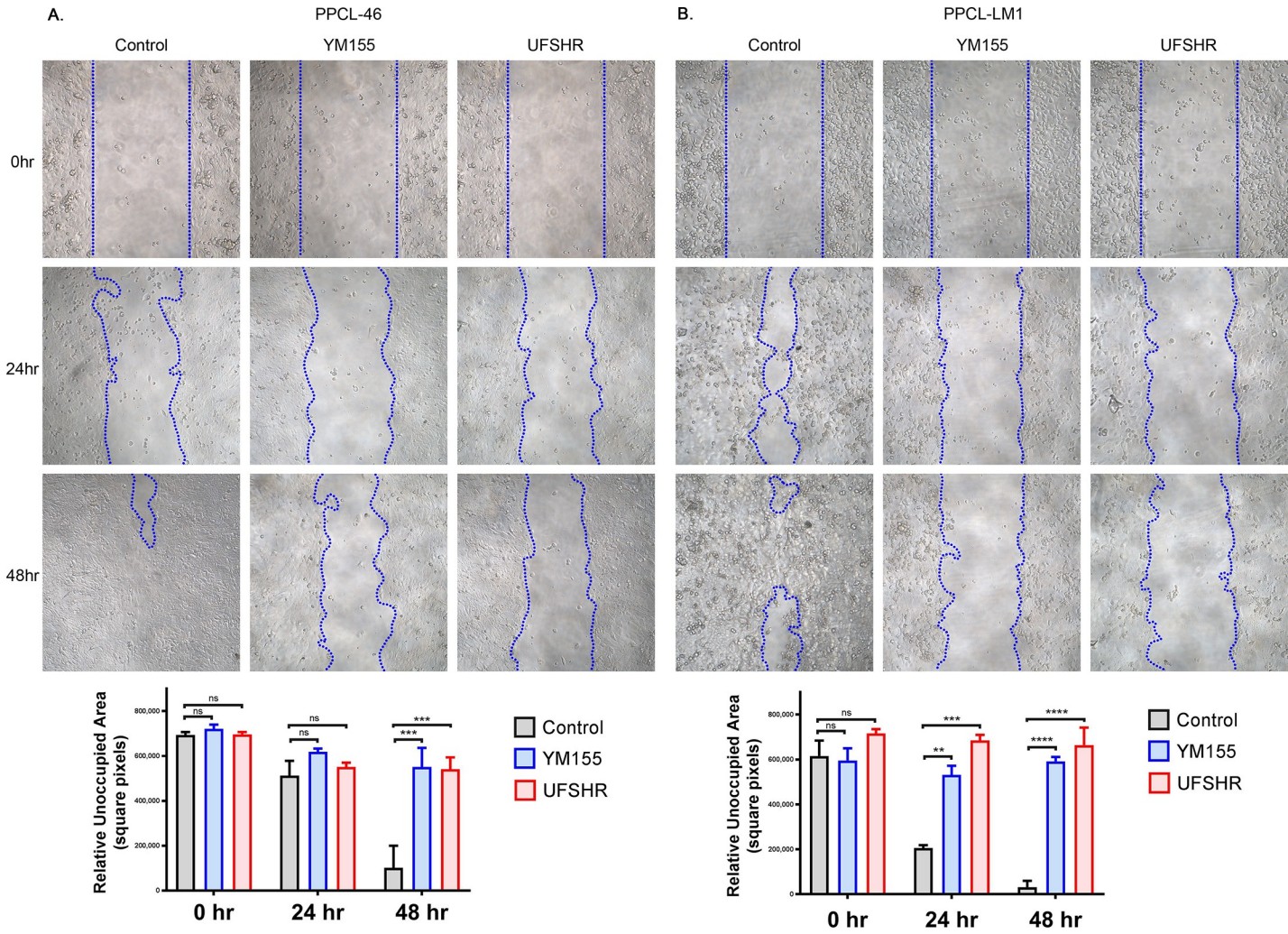

**Fig 4. Repressing survivin expression with YM155 and UFSHR reduces tumor cell migration.** Wound healing assays following treatment with YM155 (5nM) and UFSHR (50nM) at 24 and 48-hour time points in (A) PPCL-46 and (B) PPCL-LM1. Bar graphs show the calculated area of unoccupied cells using ImageJ. **P<0.01, ***P<0.001, ****P<0.0001.

observed as early as 24 hours after the treatment with the fast-migrating PPCL-LM1 (P<0.001), but not the slow-migrating PPCL-46 cells. With respect to the migratory pheno-type, there was not a significant difference in the sensitivity to UFSHR and YM155 at either time point. Altogether, our data provides solid evidence for UFSHR's activity against survivin-regulated functions of both PPCL models. Treatment with both UFHSR and YM155 reduced cell migration with a similar and significant effect at 48-hour time point. Thus, it is clear that the reduction in survivin expression following treatment with UFHSR and YM155 contributes to the reduction in tumor cell migration in PDAC.

## UFSHR significantly contributes to tumor regression in PPCL-implanted xenografts

The evaluation of UFSHR activity against tumorigenicity was further explored in an *in vivo* setting, using immuno-deficient mice implanted with PPCL-46 or PPCL-LM1 tumor cells (Fig 5A). At the end of the experiment, the tumor growth from both PPCL-46 and PPCL-LM1 was

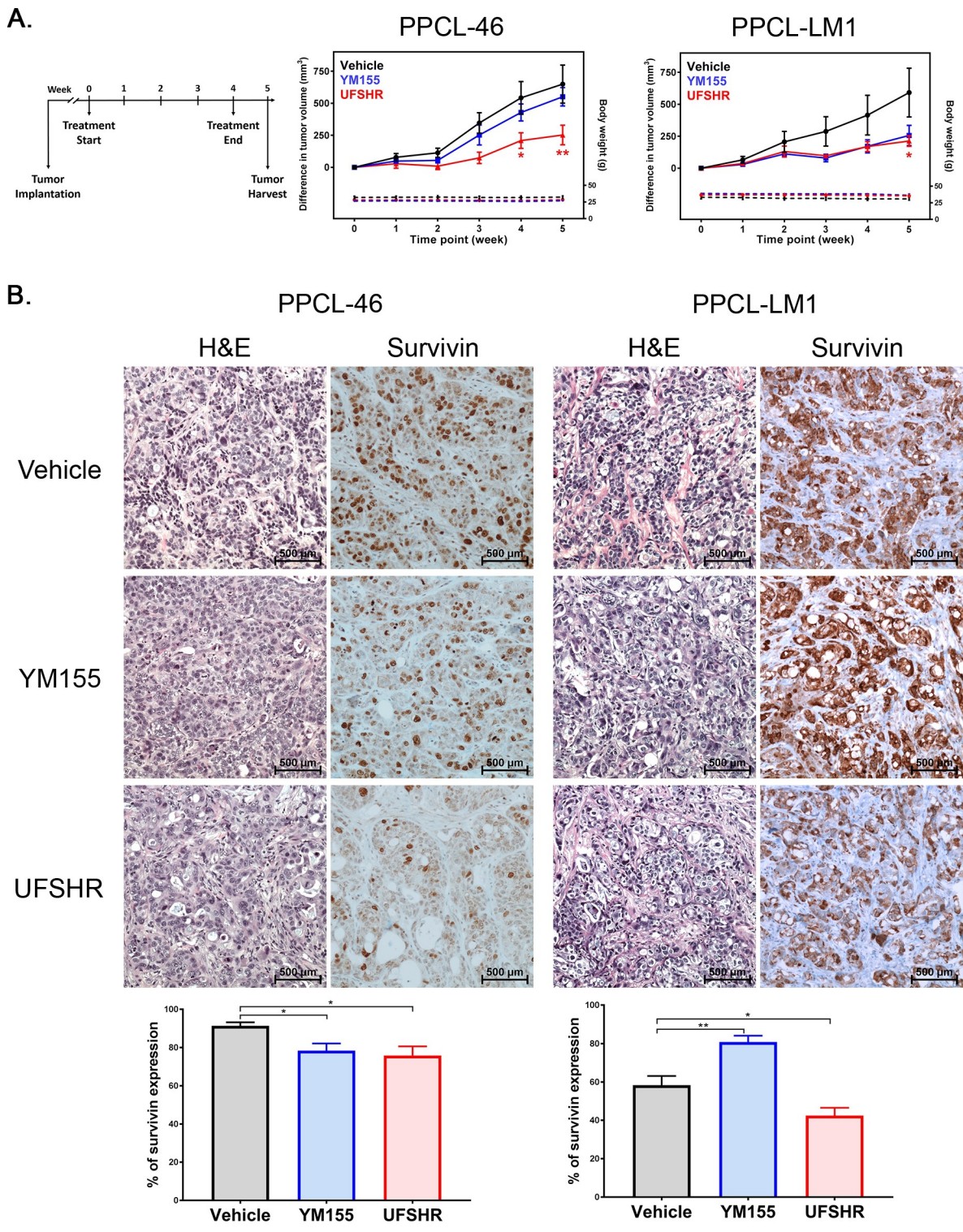

**Fig 5. Repressing survivin expression with YM155 and UFSHR *in vivo* contributes to a reduction in tumor progression.** (A) Time course of tumor cell implantation, treatment, and harvest and change in tumor volume since the commencement of treatment. (B) Hematoxylin & eosin-stained micrographs and immunohistochemical staining of survivin in PDX tumor tissue. Bar graphs represent the quantitation of survivin expression in the PDX tumor tissue.

significantly slower in the animal cohort treated with UFSHR, when compared to the vehicle group (Fig 5B). With YM155 treatment, tumor shrinkage was observed in PPCL-LM1 though it was not statistically significant. Meanwhile, there was only a subtle effect on PPCL-46 tumors following YM155 treatment. The immunohistochemical analysis showed that the tumor regression strongly correlated with survivin expression in UFSHR-treated tumors derived from both PPCLs, compared to that in the vehicle or YM155 groups (Fig 5C). In PPCL-LM1 tumors there was a statistically significant reduction in survivin expression in tumors treated with UFSHR. This *in vivo* data provides a foundation for the clinical application of UFSHR in targeting survivin and halting tumor progression with high sensitivity in the treatment of pancreatic cancer.

## Discussion

While survivin has been extensively studied in other human cancers, its contribution to the tumorigenesis of PDAC remains largely unknown. This study showed that the overexpression of survivin contributes significantly to aggressive tumor phenotypes in PDAC and that survivin has the potential to become an attractive target for PDAC treatment. Our data describes a novel small molecule inhibitor UFSHR that may provide an alternative option for targeting survivin. Albeit slightly less potent than the well-known survivin inhibitor YM155 in the *in vitro* setting, UFSHR showed a comparable effect both on the inhibition of survivin expression and the impact on the corresponding tumor phenotypes. Moreover, the significant impact on tumor progression *in vivo*, when compared to YM155 treatment, demonstrated the full therapeutic potential of this novel survivin inhibitor.

Given that survivin is overexpressed in many different types of cancer, a number of therapeutic strategies have emerged to reduce survivin expression. *In vitro* studies using pancreatic cancer cell lines have shown that knocking down survivin expression with siRNA increases their sensitivity to radiation therapy [36]. LY2181308 and SPC3042 are antisense oligonucleotides used to repress expression of survivin transcripts [37]. Preclinical studies have suggested that LY2181308 can sensitize tumor cells to cytotoxic agents such as paclitaxel or docetaxel [38]. LY2181308 has progressed to Phase II trials for advanced-stage malignancies. The drug is well tolerated; however, phase I trials have revealed limited efficacy as a monotherapy [38]. Future trials are evaluating this drug in combination with chemotherapeutic agents. SPC3042, a 16-mer locked nucleic acid oligonucleotide that exists as a phosphorothiolated gapmer with 7 LNA nucleotide flanks, has been shown to have a higher potency in terms of survivin mRNA repression in preclinical studies [39]. It remains in preclinical development, but it has been seen that the therapy contributes to increased apoptotic activity, cell cycle arrest, and reduced levels of Bcl-2. Additionally, SPC3042 sensitizes tumor cells to the chemotherapeutic Taxol in both *in vitro* and *in vivo* models [39]. In addition to antisense therapies, small molecule inhibitors have been employed to acts as therapeutic strategies targeting survivin for cancer treatment. One such is the small-molecule inhibitor YM155 described earlier. Teramceprocol (meso-tetra-O-methyl nordihydroguaiaretic acid, formerly known as EM-1421 and M4N) is another member of this drug class [40]. This therapy targets Sp1 in order to repress the transcription of survivin and Cdc2, and it is currently undergoing phase I trials for solid refractory tumors [40].

While YM155 has been successful at contributing to tumor regression, there is a variable treatment response rate being reported in early-stage clinical trials [32]. In addition, due to its lack of specificity, YM155 may not be the most ideal candidate for PDAC therapy. For example, YM155 treatment has been shown to increase the expression of BIRC8, a pro-survival gene [33]. It is also possible that DNA damage is another mechanism whereby YM155 reduces

the proliferation of malignant cells and that the silencing of survivin expression follows this event [41]. Studies in leukemia models have shown that shRNA-induced knockdown of survivin arrested cells in G2/M phase of the cell cycle whereas YM155 treatment arrested the cells in S phase [42]. Additionally, U-937 and NB4 leukemia cells lines have low survivin expression compared to that of HL-60 cells, however MTS assays showed a similar sensitivity to the drug [42]. All of this supports the claim that YM155's effect may not be due to a change in survivin expression alone.

Consistent with published data describing the translational control of survivin expression, our own RT-PCR studies showed that the YM155-analog UFHSR had a negligible impact on mRNA expression. However, UFSHR showed a potent effect on survivin protein expression in immunoblotting studies. It is possible that this YM155 analog UFSHR acts either through one of the translational control mechanisms targeted by YM155 or an entirely different mechanism of translational inhibition of survivin due to the changes in critical chemical functional groups. Therefore, this supports our future plan to conduct a comprehensive proteomic analysis in order to fully capture the mechanism of action of UFSHR on the translational control of survivin and to ascertain how this impacts downstream tumor cell signaling.

Meanwhile, our study strongly suggests the effectiveness of UFSHR, a novel analog of YM155, as an option to target survivin in order to control aggressive phenotypes in pancreatic tumor cells. In the PPCL-46 model, the prolonged sensitivity to UFSHR was significantly enhanced after 48 hours of treatment, while there is no difference in 24-hour and 48-hour response observed in YM155 treatment. The time-dependent sensitivity can account for the difference in half-life and/or stability of these inhibitors. Likewise, this may also suggest that UFSHR acts through a different mechanism from YM155 in PDAC tumor cells. Importantly, while there was not a significant difference in tumor volume following treatment with YM155, both PDX systems showed a significant impact in halting tumor progression after treatment with UFSHR at the end of the treatment cycle. This suggests that UFSHR could provide a better value for clinical translation in the treatment of pancreatic ductal carcinoma.

Translation of tailored treatment regimens on the individual patient level is exemplified by the routine practice of molecular profiling in clinical cases. PDXs provide an attractive model for personalized cancer therapy and have already shown promise in the treatment of sarcoma [43]. In this study, we used our well-established and characterized PDX models of PDAC tumors and corresponding cell lines to identify appropriate targets for therapeutic intervention and to evaluate the potential high-throughput pharmacologic screening of individualized therapeutics [34]. Although both examined PDXs exhibit a high expression of survivin, our data indicate a significant difference in the sensitivity to YM155 and UFSHR. In the *in vitro* setting, the sensitivity to survivin inhibitors in PPCL-46 models is likely time-dependent, while this is not the case in the PPCL-LM1 model. In addition, the impact of survivin inhibitors in PPCL-LM1 is mostly reflected through the enhancement of late apoptotic cells. The inhibitory effect is more profound in the early apoptotic population of PPCL-46 cells. The variation in sensitivity to the therapeutic response of these PDXs may account for the intra- and inter-tumoral heterogeneity in each individual and/or among the patient population, respectively.

The data reported herein clearly demonstrate the therapeutic potential of UFSHR in PDAC treatment. While our study indicated that neither UFSHR nor YM155 sensitized pancreatic tumor cells to the frontline chemotherapy gemcitabine, as the mechanism of these drugs become more clearly defined, there may be opportunities for combination strategies involving other targeted agents or immunotherapeutics. As an example, survivin inhibitors that induce apoptosis of cancer cells could mimic the antigen presentation seen following treatment with cytotoxic agents [44]. This antigen presentation could in turn help increase the sensitivity of pancreatic tumors to immunotherapies such as checkpoint blockade. Immunoncology

approaches with survivin have been explored in the context of PDAC. In fact, the overexpression of survivin has been a target of cancer vaccination by using survivin 2B peptides in combination with interferon-β. While this therapy did not improve progression-free survival, there was a significantly enhanced immune reaction following treatment with this vaccine [45]. Overall, continued evaluation of survivin as a therapeutic target with our novel inhibitor has the potential to provide more therapeutic options and enhance overall benefits to patients with PDAC.

## Supporting information

**S1 File. Raw immunoblot images.** Included in this file are the original, uncropped images of the western blot membranes taken at the time of exposure. In the raw images, the membranes were cut in order to probe for multiple proteins on the same membrane at the same time. An image of the protein ladder (from Bio-Rad) is included next to the raw blot image for reference. Each page of the file represents a single exposure.
(PDF)

## Author Contributions

**Conceptualization:** Chen Liu, Kien Pham.

**Formal analysis:** David Ostrov.

**Funding acquisition:** Chen Liu.

**Investigation:** Matthew Brown, Wanbin Zhang, Deyue Yan, He Wang, David Ostrov, Keith Robertson, Chen Liu, Kien Pham.

**Methodology:** Matthew Brown, Wanbin Zhang, Deyue Yan, Rajath Kenath, Le Le, He Wang, David Ostrov, Keith Robertson, Kien Pham.

**Project administration:** Chen Liu, Kien Pham.

**Resources:** Wanbin Zhang, Deyue Yan, He Wang, Daniel Delitto, Keith Robertson, Chen Liu.

**Software:** David Ostrov.

**Supervision:** Chen Liu, Kien Pham.

**Validation:** He Wang.

**Writing – original draft:** Matthew Brown, Kien Pham.

**Writing – review & editing:** Matthew Brown, Chen Liu, Kien Pham.

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
