## [Decision Letter · Decision Letter 0]

2 Sep 2019

PONE-D-19-19723

The Role of Survivin in the Progression of Pancreatic Ductal Adenocarcinoma (PDAC) and a Novel Survivin-Targeted Therapeutic for PDAC

PLOS ONE

Dear Dr Pham,

Thank you for submitting your manuscript to PLOS ONE. After careful consideration, we feel that it has merit but does not fully meet PLOS ONE’s publication criteria as it currently stands. Therefore, we invite you to submit a revised version of the manuscript that addresses the points raised during the review process.

We would appreciate receiving your revised manuscript by Oct 17 2019 11:59PM. To enhance the reproducibility of your results, we recommend that if applicable you deposit your laboratory protocols in protocols.io, where a protocol can be assigned its own identifier (DOI) such that it can be cited independently in the future. For instructions see: http://journals.plos.org/plosone/s/submission-guidelines#loc-laboratory-protocols

We look forward to receiving your revised manuscript.

Kind regards,

Andrei L. Gartel

Academic Editor

PLOS ONE

Journal Requirements:

2. In your Methods section, please provide additional information about the pancreatic tumor samples used to derive the PDX models. Please ensure you have provided sufficient details to replicate the analyses such as: a) the date range (month and year) during which patient samples were collected, b) a description of relevant demographic details (sex, stage, molecular subtype) and c) the number of independent samples used in this work.

Additional Editor Comments (if provided):

Please answer all questions from the reviewer.

Reviewers' comments:

Reviewer's Responses to Questions

**Comments to the Author**

1. Is the manuscript technically sound, and do the data support the conclusions?

Reviewer #1: Yes

2. Has the statistical analysis been performed appropriately and rigorously? 

Reviewer #1: No

3. Have the authors made all data underlying the findings in their manuscript fully available?

Reviewer #1: Yes

4. Is the manuscript presented in an intelligible fashion and written in standard English?

Reviewer #1: Yes

5. Review Comments to the Author

Reviewer #1: The authors present data validating survivin as a target in pancreatic adenocarcinoma using TCGA data and histologic evaluation of primary tissues compared to histologic normal tissue. There have been earlier publications showing survivin is upregulated in pancreatic cancer (Satoh. Cancer 2001;92:271–278) and its inhibition using knockdown sensitizes pancreatic CA to radiation (Kami. Surgery 2005;138:299–305).

The authors use unique patient derived cell line models-one is from the primary cancer and other is derived from a metastatic pancreatic cancer lesion. It would be interesting to see at what stage in transformation this protein is upregulated in pancreatic adenoCA, looking at the continuum of adenomas, in situ and invasive carcinoma-primary or metastatic lesions. Moreover with these 2 different cell lines differences seen in the efficacy of the survivin inhibition may be related to differences in the relevance of this protein in primary versus metastatic lesion.

Another novelty of this research is the use of UFSHR a novel derivative of YM155 which is a known survivin inhibitor by destabilizing the transcriptional complex at the BIRC promoter.

However the authors have not confirmed the mechanism by which this drug is inhibiting survivin –there is no examination of survivin gene expression to confirm transcriptional repression of BIRC5. Are there differences in BIRC8 which is upregulated by YM155as an off target effect? Do the 2 drugs differ in their effects on survivin splice variants that are responsible for the anti-apoptotic effects of this gene

The concept of inhibiting survivin in this cancer is not novel- a peptide survivin vaccine in combination with interferon in a phase 1 trial in pancreatic cancer was published in Cancer Science in 2018. The PDX animal data is compelling. This drug is inhibiting tumor growth and suppressing the target in vivo. Whether survivin is critical to this anticancer activity of UFSHR should be proven by overexpressing survivin to reverse the apoptotic phenotype because destabilizing a transcriptional complex could have multiple effects.

Statistics have not been shown to justify the size of the animal experiments nor is there any mention of any statistical tests used in the in vitro experiments.

6. PLOS authors have the option to publish the peer review history of their article (what does this mean?). If published, this will include your full peer review and any attached files.

Reviewer #1: No

---

## [Author Response · Author response to Decision Letter 0]

31 Oct 2019

To Reviewer and Editor,

On behalf of all authors, I would like to thank the Reviewer and Editor for your thoughtful comments. We have made every effort to address these comments point to point. We hope that the changes that we made in the revised version will meet your satisfaction. 

Best regards, 

Kien Pham

---

## [Editor Report · Decision Letter 1]

10 Dec 2019

The Role of Survivin in the Progression of Pancreatic Ductal Adenocarcinoma (PDAC) and a Novel Survivin-Targeted Therapeutic for PDAC

PONE-D-19-19723R1

Dear Dr. Pham,

We are pleased to inform you that your manuscript has been judged scientifically suitable for publication and will be formally accepted for publication once it complies with all outstanding technical requirements.

With kind regards,

Andrei L. Gartel

Academic Editor

PLOS ONE
---

## [Editor Report · Acceptance letter]

30 Dec 2019

PONE-D-19-19723R1 

The Role of Survivin in the Progression of Pancreatic Ductal Adenocarcinoma (PDAC) and a Novel Survivin-Targeted Therapeutic for PDAC 

Dear Dr. Pham:

I am pleased to inform you that your manuscript has been deemed suitable for publication in PLOS ONE. Congratulations! Your manuscript is now with our production department. 

With kind regards,

on behalf of

Dr. Andrei L. Gartel 

Academic Editor

PLOS ONE